# Physiological Response of *Pelophylax nigromaculatus* Adults to Salinity Exposure

**DOI:** 10.3390/ani10091698

**Published:** 2020-09-20

**Authors:** Jun-Kyu Park, Yuno Do

**Affiliations:** Department of Biological Sciences, Kongju National University, Gongju 32588, Korea; pjk8578@smail.kongju.ac.kr

**Keywords:** freshwater ecosystems, permeable skin, physiological response, recovery, renal function, salinity changes

## Abstract

**Simple Summary:**

This study explored physiological resilience to and recovery from saline exposure in *Pelophylax nigromaculatus*, a semi-aquatic frog that is widely distributed in East Asia. Analysis of 11 serum components revealed the physiological response of frogs to either severe saline exposure for six days or moderate saline exposure for forty days, followed by a twenty day recovery period. During exposure to both severe and moderate saline conditions, serum electrolytes increased, protein concentrations in serum decreased, and creatinine, an indicator of renal function, sharply increased. However, renal tissue sampled after the study did not show renal dysfunction. Moreover, serum components that changed during exposure to salinity returned to their initial values during the recovery period. Adult anurans seem capable of resilience, to some extent, to saline conditions.

**Abstract:**

Many freshwater ecosystems are becoming more saline, and amphibians, which have permeable skin, are sensitive to this change. We studied the physiological responses to high salinity and recovery from saline exposure in adult frogs (*Pelophylax nigromaculatus*). Frogs that experienced severe salinity were exposed to saline conditions for 6 days, while those in the moderate group were exposed to saline conditions for 40 days, followed by a recovery period in freshwater for 20 days. Our data showed that during exposure to saline conditions of severe and moderate groups, serum electrolytes increased, protein concentrations decreased, and creatinine, an indicator of renal function, sharply increased. However, renal tissue sampled after exposure did not show renal dysfunction. In addition, serum components that changed during exposure to salinity returned to their initial values during the recovery period. Thus, adult anurans can be resilient, to some extent, to saline conditions in habitats that experience either rapid or slow salinity changes.

## 1. Introduction

Freshwater ecosystems have been degraded by increases in salinity [1], and many organisms inhabiting in freshwater habitats are physiologically affected by this salinization [2]. In particular, amphibians are more sensitive to changes in salinity than other taxa [3] because they have permeable skin and a complex life history [4]. Amphibians are highly threatened, experiencing the highest extinction rates in vertebrates (ca. 40%) due to various factors, but most of these factors (ca. 48%) are “enigmatic factors” [5]. The enigmatic threaten factors of amphibians may include changes in salinity in freshwater habitats. The changes in salinity have various effects on the amphibian population [6,7,8], and can actually be frequently observed in them. More than 100 amphibian species have been found in habitats that are exposed to predictable saline conditions, such as mangrove forests that experience tidal cycles or habitats that undergo high rates of evaporation during certain periods [9,10]. They have also been found in habitats with unpredictable salinity, such as areas affected by sea-level rise due to climate change, areas that experience seawater intrusion during coastal storms, intermittently opening estuaries, and habitats affected by salt deicing treatments from roads [11,12,13,14]. We need to understand the physiological response of amphibians in all life history by change of salinity to come up with ways to efficiently protect and manage them and to identify how they survive and adapt in predictable or unpredictable saline habitats.

At each stage of their life history, frogs have a varying level of response to salinity exposure. Female frogs avoid spawning sites with high salinity [15]. Tadpoles increase the expression of Na+-K+ ATPase (NKA) in their gills to oppose the influx of ions by increased salinity [16]. They also have a compensatory growth system that activates once conditions improve, to minimize the negative effects of salinity experienced early in development [17,18]. Both adults and tadpoles may become less sensitive to salinity changes by gradually increasing their salt tolerance when they experience predictable salinity exposure or show acclimation in moderate salinity [16,19]. The relative degree of resilience obviously depends on the stressor factor and its intensity [20,21,22]. Frogs may have adequate physiologic reserves to recover from moderate and predictable salinity exposure yet lack sufficient resilience to recover from severe and unpredictable salinity exposure. Generally, high saline conditions affect corticosterone in adult anurans along the hypothalamic–pituitary–adrenal axis, acting as a stress [23]. However, moderate salinity can reduce the rate of spread and infection of bacteria and parasites in anurans [24,25]. Frogs infected with bacteria or parasites often prefer saline habitats over freshwater [26]. Adult frogs tolerate salt by absorbing ions and by synthesizing and retaining urea [27,28], while tadpoles increase the osmotic pressure in their bodies through ion absorption alone. Although several studies have investigated salt tolerance in adult frogs, no study has explored whether changes in frogs’ physiology return to normal levels after cessation of salinity exposure. This resilience is one of the features of amphibians that has enabled them to survive mass extinction events and successfully adapt to extreme habitats [29]. The present study explores how adult amphibians can use saline habitats by measuring their response to salinity and recovery from saline exposure to understand their adaptation to changing habitats.

We hypothesized that adult frogs suffer temporary negative effects when they are exposed to increasing salinity, but they can quickly recover when normal conditions are restored. We examined the salinity response and recovery of adult black spotted pond frogs (*Pelophylax nigromaculatus*), a semi-aquatic pond species that is widely distributed in East Asia across China, Japan, and Korea [30]. Serum chemistry was used to diagnose physiological changes, and renal tissues were inspected to determine how the kidneys, which function in homeostasis, were affected by salinity. The goals of the study were to identify: (i) how salinity changes the physiological status and the body condition of adult frogs, and (ii) whether the physiological status and body condition return to normal after normal environmental conditions are restored. A better understanding of how adult anurans respond to salinity and recover from exposure will shed light on the distribution and survival patterns of amphibians in coastal habitats and habitats with potentially changing saline conditions.

## 2. Materials and Methods

### 2.1. Experimental Animals

The study animals were 34 *P. nigromaculatus* adults collected from a rice paddy in Gongju, Chungcheongnam-do, South Korea, during the breeding season in 2019. The average snout–vent length was 68.3 ± 7.1 mm, and average weight was 37.6 ± 11.5 g (mean ± SD). We measured salinity in collection site at three different times using a multi parameter water quality meter (YSI Pro-Plus, Yellow Springs, OH, USA), and confirmed that the salinity of the habitat was around 0.14 (Appendix A). Frogs were transported to the lab in plastic boxes containing water collected from the rice paddy. In the lab, frogs were acclimated over 72-h to the study conditions in covered plastic aquaria (460 mm × 300 mm × 170 mm) with a bed of sterile coconut fiber and free access to a water pool containing 2 L of bottled water under a 12-h/12-h light/dark regime with Exo-tera UVB 100 lamps (UVB 30%, UVA 5%) and air temperature 25 ° ± 2 °C. Animal maintenance and experimental procedures were in accordance with the regulations and with the approval of the Experimental Animal Ethics Committee of Kongju National University (KNU_2019-01).

### 2.2. Experimental Design and Treatment

We set up two experimental conditions: a severe and a moderate exposure (Figure 1). In the first case (severe condition), frogs were bred in closed rectangular plastic containers (460 mm × 300 mm × 170 mm) with a water pool but no bed of fiber. Under the severe condition, frogs remained in the water during the entire experimental period and could not escape the water. Six frogs were divided into an experimental group (three individuals) and an control group (three individuals). In a preliminary test, the serum sodium and chlorine of frogs under severe conditions were significantly changed on 3 days to 6 days, and changed serum components were maintained since then. Therefore, frogs in the experimental groups were exposed to 5‰ saline (specific conductivity; about 9400 μS/cm, ion conductivity; about 8700 μS/cm) water for six days, while frogs in the control groups were maintained in freshwater below 0.4 ‰ salinity (specific conductivity; about 350 μS/cm, ion conductivity; about 340 μS/cm) for six days.

In the second case (moderate condition), frogs were bred in plastic containers (460 mm × 300 mm × 170 mm) with a water pool and the bed of fiber. Frogs had free access to land (on the bed of fiber) and water (in the pool) in moderate conditions. A total of 10 frogs were divided into an experimental group (five individuals) and a control group (five individuals). In a preliminary test, the serum sodium and chlorine of frogs under moderate conditions were significantly changed from 30 days to 40 days and changed serum components were maintained from then onwards. Frogs of the experimental group were exposed to 5‰ (specific conductivity; about 9000 μS/cm, ion conductivity; about 8700 μS/cm) saline water for 40 days (the salinity treatment period). After salinity exposure, frogs were kept in a container with freshwater less than 0.4‰ (specific conductivity about 510 μS/cm, ion conductivity about 450 μS/cm) salinity for 20 days (the recovery period). The other five frogs in the control group were raised with less than 0.4 ‰ (specific conductivity; about 590 μS/cm, ion conductivity; about 560 μS/cm) of bottled water for 60 days.

Each frog was housed in a container individually. We fed frogs with mealworms (ca. 400 mg) and crickets (ca. 400 mg) with Rep-Cal herptivite calcium (vitamin D3 and phosphorus 0%) and Rep-Cal herptivite multivitamins (with beta carotene) powder every third day. All of the water and coconut fiber were also cleaned every 10 days. We measured salinity and water quality in all water of containers on each blood extraction (Appendix A). During the experimental period, air temperature in all containers was maintained at 25 ° ± 2 °C with a 12-h/12-h light/dark regime with Exo-tera UVB 100 lamps (UVB 30%, UVA 5%).

### 2.3. Blood Samples and Serum Chemistry Analysis

Blood samples (below 0.4 mL or equivalent to 1% of weight) were collected from anesthetized frogs in ice-cold water by cardiac venipuncture using a heparinized syringe. Blood samples from six frogs experiencing severe conditions were collected every third day. In the moderate conditions, blood samples from 10 frogs were collected every 10 days (Figure 1). To establish a reference interval for serum components, we also collected blood samples from 18 frogs that were not used in the experiment. Blood samples collected from 16 experimental frogs immediately after acclimation were also used as a reference interval for serum components.

A blood biochemistry analyzer (Beckman coulter AU680, Miami, FL, USA) was used to analyze three electrolyte components: sodium, potassium, and chlorine. A discrete-type clinical chemistry automated analyzer (Hitachi Automatic Analyzer 7020, Tokyo, Japan) was used to analyze eight serum components: total protein, albumin, total globulin, glucose, alanine aminotransferase (ALT), aspartate aminotransferase (AST), blood urea nitrogen (BUN), and creatinine.

The electrolyte components measured the change in osmotic state due to the stressful saline exposure. The levels of total protein, albumin, and total globulin represent homeostasis and indicate renal function [31]. Glucose levels indicated the individual’s metabolic and nutritional status. ALT and AST indicated liver function as a response to salinity. ALT is a liver-specific indicator, but it is so sensitive to changes that it is difficult to determine the functional state of the liver from elevation of this enzyme alone. Therefore, AST, which increases with damage to the liver, heart, and skeletal muscle [31], was also measured. BUN and creatinine were analyzed to diagnose renal function abnormalities due to changes in salinity. Because BUN is a waste product from protein metabolism, it can reflect protein metabolism in addition to elevation of renal stress [32]. Creatinine is an indicator of renal function that specifically suggests renal dysfunction when it is elevated alone [32]. The mean, standard error of measurement (SEM), 95% confidence interval, 25th/75th percentile, and median as reference interval of *P. nigromaculatus* were calculated for the electrolyte and serum components (Table 1).

### 2.4. Histology

After the experimental period was over, six frogs from server conditions (3 frogs of the experimental group and 3 frogs of the control group) and ten frogs from the moderate conditions (5 frogs of the experimental groups and 5 frogs of the control groups) were euthanized by pithing, and renal tissue of frogs were rapidly collected and fixed in 10% formalin solution. A fixed sample of renal tissue was cut to a thickness of 2–3 mm, and was washed in deionized water. The paraffin embedding of samples was carried out by using a Spin Tissue Processor STP 120 (Myr, Tarragona, Spain) in three steps: (1) a dehydration process using ethanol (2) a clearing process using xylene (3) a paraffin embedding process. Thermo-Shandon Finesse ME Microtomes (Thermo Fisher Scientific, Waltham, MA, USA) were used to make the tissue section created by cutting to 3 µL thickness. The resulting tissue section was attached to a slide glass, and washed with deionized water after section was dried, was deparaffinized, and was hydration process. Afterwards, the following Harris’s hematoxylin and eosin staining process was performed: (1) the samples were stained with hematoxylin for 10 min, and the stained samples were washed with running water for 3 min (2) after staining the samples with the eosin solution for 1 min and 40 s, a 4-step hydration process was performed for 1 min (3) clearing was performed for 3 min using xylene, and after fixing the samples using the glycerol, the cover glass was covered and enclosed. The glomerulus and tubules of the renal system were examined to identify renal failure, and abnormalities due to saline exposure were examined using a light microscope with a camera (Olympus DP72, Tokyo, Japan) at 100× magnification. In the case of the glomerulus, a predominance of polymorphonuclear leukocytes and glomerular congestion are observed when there is a histological renal dysfunction. It can also appear as patches and expansion of the glomerular mesangium. In tubules with abnormal renal function, the reactive and degenerative nuclear changes, intraluminal eosinophilic material, remaining nuclei with prominent nucleolus, acute tubular necrosis etc. can be observed, represented by the shape of the nucleus or deformation of the tubular edge. It also develops a tumor form in which the meninges of the tubules swell [33]. We confirmed the histological renal dysfunction of treatment groups from severe and moderate conditions through comparison with the renal images of control groups from severe and moderate conditions.

### 2.5. Statistical Analysis

The means ± SEM were calculated for the variables measured. Two-way repeated measures analysis of variance (ANOVA) were used to assess both the effect of time (repeated measurements of the same individual) and the effect of treatment. Where a significant difference occurred, the Tukey’s post hoc test were performed. A value of *p* < 0.05 was considered significant. All analyses were performed using GraphPad Prism version 7.0 for Windows (GraphPad Software, San Diego, CA, USA).

## 3. Results

### 3.1. Response to the Severe Condition

In the control group of severe salinity conditions, blood electrolytes (sodium, chlorine, and potassium, Figure 2a–c), did not change after 6 days (Tukey’s test, *p* > 0.05). Blood protein (total protein, albumin, and total globulin, Figure 2d–f) of the control group sharply decreased after 3 days (*p* < 0.05). Glucose sharply decreased (*p* < 0.05) from 0 days to 3 days and increased (*p* < 0.05) from 3 days to 6 days (Figure 2g). ALT and AST, which are liver function indicators, were not changed after 6 days (*p* > 0.05, Figure 2h,i). BUN, which represents protein metabolism, sharply increased from 3 days to 6 days (*p* < 0.05, Figure 2j), while Creatinine, which is the renal function indicator, did not change after 6 days (*p* > 0.05, Figure 2k).

Under the experimental groups of severe salinity condition, sodium and chlorine levels sharply increased over the course of 6 days. The levels of two blood electrolytes were significantly higher than the initial level after 3 days (*p* < 0.05, Figure 2a–c). Total protein and albumin steadily decreased (*p* < 0.05) after 3 days, while total globulin did not significantly change after 6 days of salinity exposure (*p* > 0.05, Figure 2d–f). Glucose did not change (*p* > 0.05, Figure 2g). ALT and AST, which are indicators of liver function, did not change (*p* > 0.05, Figure 2h,i). BUN sharply decreased after 3 days (*p* < 0.05, Figure 2j). Creatinine, an indicator of renal function, sharply increased after 6 days (*p* < 0.05, Figure 2k).

### 3.2. Response to the Moderate Condition

In the control group, none of the serum electrolytes changed from days 0 to 60 (*p* > 0.05, Figure 3a–c). The total protein, albumin, and total globulin sharply increased from 0 to 10 days (*p* < 0.05) and remained higher than initial levels (*p* > 0.05, Figure 3d–f). Glucose rapidly increased from 0 to 20 days (*p* < 0.05) and then sharply decreased from 20 to 30 days (*p* < 0.05, Figure 3g). ALT and AST did not change from 0 to 60 days (*p* > 0.05, Figure 3h,i). BUN was lower at 10 days than it was initially (*p* < 0.05) and did not change further from 10 to 60 days (*p* > 0.05, Figure 3j). Creatinine was not change from 0 to 60 days (*p* > 0.05, Figure 3k).

Under the moderate salinity condition, two blood electrolytes, sodium and chlorine, steadily increased in the experimental group from 0 to 40 days (*p* < 0.05), while potassium was unchanged during the same period (*p* > 0.05). After the saline exposure was discontinued, sodium and chlorine gradually decreased (*p* > 0.05) and returned to a level that was not significantly different from the initial level (*p* > 0.05, Figure 3a–c).

The total protein, albumin, and total globulin of blood serum decreased to their lowest levels in 40 days (*p* < 0.05). The levels of these three blood proteins increased after salinity exposure was discontinued (*p* < 0.05). Total protein, albumin, and total globulin recovered in a similar fashion to the initial levels (*p* > 0.05, Figure 3d–f).

The glucose level remained low, similar to initial levels (*p* > 0.05). In experimental groups, glucose in blood serum was not changed (*p* > 0.05) during salinity treatment and during the recovery period (Figure 3g).

ALT and AST did not change from 0 to 60 days during saline exposure and the recovery period (*p* > 0.05, Figure 3h,i). BUN was lower at 10 days than it was initially (*p* < 0.05) and did not change further from 10 to 60 days (*p* > 0.05, Figure 3j). Creatinine sharply increased during salinity exposure and peaked at 40 days (*p* < 0.05, Figure 3k). During the recovery period, creatinine decreased again from 50 to 60 days (*p* > 0.05) and recovered to initial levels by day 60 (*p* > 0.05).

### 3.3. Renal Tissue Inspection

Renal tissue was analyzed for the normality of tubules and glomerulus at four random sites in both kidneys. When the three renal images of the treatment group in the severe group were compared with the three renal images of the control group, evidence of histological renal abnormalities (such as patches and expansion of the glomerular mesangium, reactive and degenerative nuclear changes, intraluminal eosinophilic material, remaining nuclei with prominent nucleolus, acute tubular necrosis, deformation of the tubular edge etc.) were not found in the renal images of frogs in the treatment group (Figure 4a,b). Under the moderate condition, the five renal images of the treatment groups did not present histological renal abnormalities when the renal images were compared with the five images of the control groups (Figure 4c,d).

In addition, ascites in two frogs of the experimental group from the severe conditions was confirmed at 6 days, whereas ascites could not be identified in all other frogs in the experimental and control groups from severe conditions and moderate conditions (Figure 5). We summarize the process of salinity response and recovery in adult frogs in Figure 6.

## 4. Discussion

Our data revealed the response of salinity exposure and recovery in adult frogs. Blood electrolytes, protein levels, and renal function levels of frogs were changed by salinity exposure under both severe and moderate conditions. The serum components returned to their initial values after saline exposure was discontinued. Histological examination revealed no abnormalities or disease of the kidneys.

Previous studies showed that total protein in tadpoles decreased as water salinity increased. These changes are due to changes in the metabolic rate or energy allocation associated with osmotic maintenance [34]. However, in our study, the decreased total protein and albumin with saline exposure under both severe and moderate conditions is likely to be the result of increased protein emissions from the kidneys. Blood proteins commonly decrease due to (1) increased protein emissions from the kidneys, indicating problems of renal function, (2) reduced production of proteins due to degradation of liver function, or (3) increased protein metabolisms, as protein is converted to glucose in the liver [35]. In our study, blood sodium, chlorine, and creatinine increased with salinity exposure, and both sodium and creatinine exceeded the reference interval during the same period. Increased levels of sodium and chlorine in the blood are known to increase the glomerular filtration rate, which may play a role in kidney damage or functional degradation [36,37]. Another proof is that protein metabolism, as measured by the BUN level, declined and the liver function indicators AST and ALT were not changed by saline exposure. As decreased protein levels in experimental groups do not appear to be due to degraded liver function or increased protein metabolism, they must be due to problems in renal function caused by increased osmotic pressure. In contrast, the reduction of proteins from 3 days to 6 days in the control group that experienced the severe condition seem to be associated with protein metabolism. The decline in glucose after 3 days, increase in glucose after 6 days, and increased BUN can support the explanation for the reduction of proteins by protein metabolism.

In the control group that experienced moderate conditions, glucose dramatically decreased at 20 days, and this appears to be related to creatinine exceeding the reference interval at 30 days. Creatinine produced in muscle increases during muscle metabolism because creatine is used as an energy source [38,39]. In this process, glucose in the blood is used as an energy source to move the muscles and help convert ADP to ATP [38,40]. Unlike the control groups under both severe and moderate conditions, unchanged glucose present in both the treatment groups suggests that increased creatinine and decreased blood protein indicate damaged kidney function. Unlike our study, in a previous study, glucose levels in tadpoles that were exposed to saline conditions were significantly lower than in the control group tadpoles [34]. Mitochondrion-rich cells (MRCs), which use energy for active transport, are concentrated in the gills of tadpoles [41]. Lower levels of glucose and decreased protein in tadpoles that were exposed to saline conditions are likely due to the use of the active transport pump to move sodium and potassium ions, which is a result of the expression of NKA in the gills, a key site for ion transport [42]. In contrast, the change in blood electrolytes in adult frogs due to saline exposure seems to result from the passive transport pump of sodium and chlorine ions. Blood sodium and chlorine in amphibians can be affected by passive transport pumps in the skin and muscles, as well as by the active transport pump [43]. In fact, expression of NKA in the skin and muscle tissues of adult amphibians in habitats with relatively high salinity was no different than that of adult amphibians in freshwater [23]. This suggests that the uptake of ions in adult amphibians can be achieved by passive transport, without using energy. Our results show that tadpoles and adult amphibians use different salt tolerance mechanisms.

Amphibians are sensitive to high osmotic solutions such as saline water and to dehydration because they have poor osmoregulatory capacity due to their permeable skin and dependence on both terrestrial and aquatic habitats. Salt tolerance and resilience, as well as dehydration resistance, can determine species differentiation and habitat use and can have important consequences for species adaptation and evolution [4,44]. Some researchers support the hypothesis that dehydration and salt tolerance may have evolved together in amphibians [45,46], because green toads (*Bufotes viridis*) found in both arid and saline environments use the same physiological mechanism to synthesize and maintain the urea to survive in both habitats [27,47]. However, other researchers believe that dehydration and salt tolerance independently evolved in amphibians, arguing that there is no significant interaction between the capacities to tolerate the dehydration and salt conditions, and that there may be no genetic association between them [48]. In addition, the fundamental difference in the osmotic mechanism pre- and post- metamorphosis supports the hypothesis that tolerance of dehydration and salinity have different origins [4,34,48,49]. We concur with this view; our data suggest that the adaptation mechanism in larvae that use NKA for salt tolerance and in adult amphibians may be different because adult anurans do not have a mechanism for salt tolerance using glucose. In addition, the physiological mechanism of dehydration as described in previous studies seems to differ from the response to saline conditions in the present study. Dehydration tolerance in amphibians is well known to use physiological mechanisms that maintain blood volume by replenishing solution from the lymphatic system and by increasing water resorption in the kidneys [44]. Our results show that the salinity tolerance in adult anurans can be achieved by increasing osmotic pressure through automatic ion concentration. We believe that these mechanisms are different and evolved independently.

Although the reaction to salinity occurred faster in the group exposed to severe conditions, the groups exposed to moderate conditions also experienced a response to salinity. Furthermore, although eggs and larvae are known to be more sensitive to salinity than adult amphibians, our data show that adult amphibians can also suffer from exposure to saline conditions. As adult frogs use both terrestrial and aquatic habitats, they cannot avoid water, and react to saline conditions in the water even when they have a dry-land refuge. Our results clearly show that salinity causes a response, including physiological damage, in adult amphibians.

When the saline exposure was recovered, altered blood electrolytes and proteins rapidly returned to normal levels by 50 days, and creatinine returned to normal by 60 days. In both treatment and control groups under severe and moderate conditions, kidney biopsies revealed no glomerular or tubular abnormalities, and there were no signs of renal disease or kidney dysfunction. Exposure to 5 ‰ salinity increased osmotic pressure and temporarily degraded renal function in this species, leading to increased release of protein from the kidneys. However, this response did not seem to cause renal disease or kidney dysfunction. As a result, adult frogs can be exist in saline habitats, as shown by their ability to recover normal functioning when saline conditions were recovered. This suggests that adult frogs can live in habitats with unpredictably changing salinity and can recover rapidly when salinity returns to normal due to natural factors such as rainfall. However, salinization in some habitats with unpredictably changing salinity may be occurring much faster than some amphibian species can adapt [50]. This can have various impacts on population size, genetic diversity and flow, and stress interactions. This phenomenon does not necessarily preclude amphibian survival in these habitats, and populations may persist there due to their own physiological resilience and salt tolerance [4]. Providing species that are suffering from habitat degradation due to persistent or temporary salinity increases with access to freshwater ponds can mitigate the stress or damage they experience at all life history stages.

## 5. Conclusions

Amphibians exist in saline habitats by having salt tolerance and physiological resilience to saline conditions. When blood electrolytes increase above a certain level, they seem to cause problems of renal function in anurans. Adult amphibians are sensitive to increased salinity and experience temporary physiological changes, but when saline conditions return to normal, they show physiological resilience and quickly recover. Even adult amphibians that use terrestrial habitats experience responses to saline conditions, because they are also dependent on aquatic environments, but the stress and damage they experience can be reduced if they have access to temporary ponds containing freshwater. Our results suggest that individuals can adapt to environments with both slowly and rapidly changing salinity due to climate change or artificial factors. Their physiological changes can help us to understand the positive or negative effects of salinity involved in climate change, immunity, and stress, which can assist species conservation efforts.

## Figures and Tables

**Figure 1 animals-10-01698-f001:**
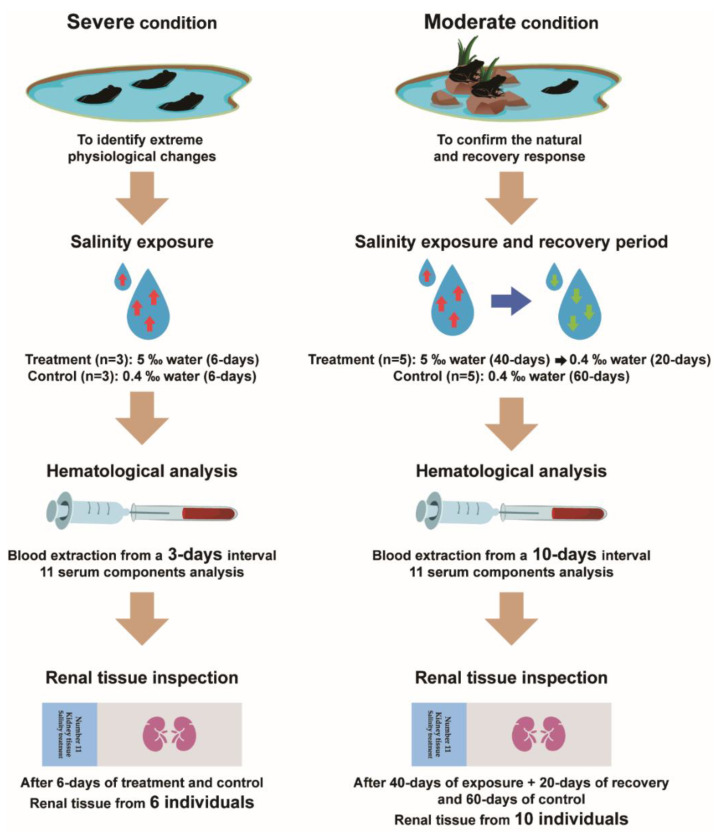
Flow chart for the present study. Wild-caught frogs were raised in treatment and control groups exposed to two conditions, moderate and severe saline exposure. In both groups, blood serum chemistry analysis and renal tissue inspection were used to identify physiological changes in frogs.

**Figure 2 animals-10-01698-f002:**
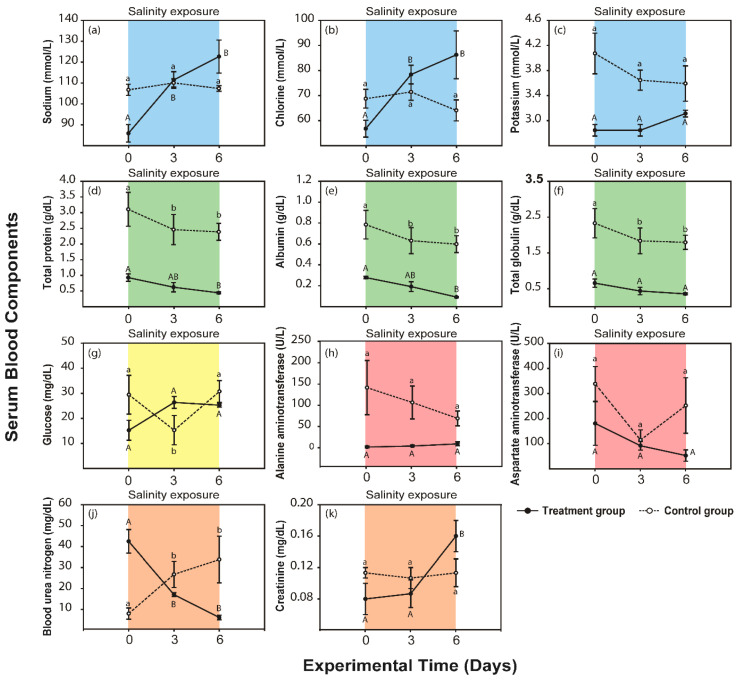
Graphs of 11 blood serum component levels (means ± standard error (SE)) after salinity exposure under the severe condition (black lines and points) and in the control group (dotted lines and white points) in *P. nigromaculatus*. The blue graphs depict the changes in serum electrolytes: (**a**) sodium (mmol/L) (**b**) chlorine (mmol/L), and (**c**) potassium (mmol/L). The green graphs depict the changes in serum protein: (**d**) total protein (g/dL), (**e**) albumin (g/dL), and (**f**) total globulin (g/dL). The yellow graph represents the change in a serum nutrient, (**g**) glucose (mg/dL). The red graphs depict the changes in liver function according to (**h**) alanine aminotransferase (ALT, U/L) and (**i**) aspartate aminotransferase (AST, U/L). The orange graphs depict the changes in renal function according to (**j**) blood urea nitrogen (BUN, mg/dL) and (**k**) creatinine (mg/dL). Significant differences (*p* < 0.05) are indicated by capital letters in the experimental group and by lowercase letters in the control group according to Tukey’s post hoc tests.

**Figure 3 animals-10-01698-f003:**
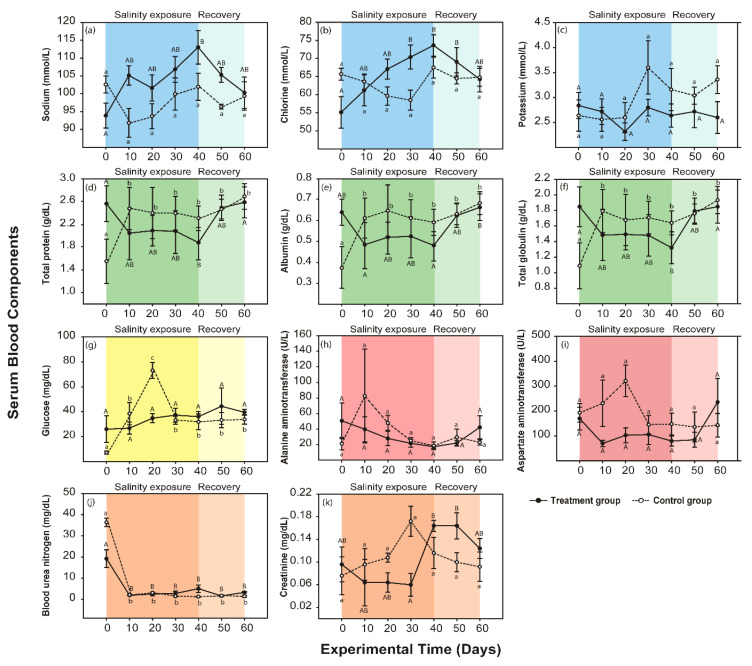
Graphs of 11 blood serum component levels (means ± SE) after salinity exposure under the moderate condition (black lines and points) and in the control group (dotted lines and white points) in *P. nigromaculatus*. The dark color indicates a treatment period, while the relatively light color indicates a recovery period. The blue graphs depict the changes in serum electrolytes: (**a**) sodium (mmol/L), (**b**) chlorine (mmol/L), and (**c**) potassium (mmol/L). The green graphs depict the changes in serum protein: (**d**) total protein (g/dL), (**e**) albumin (g/dL), and (**f**) total globulin (g/dL). The yellow graph represents the change in a serum nutrient, (**g**) glucose (mg/dL). The red graphs depict the changes in liver function according to (**h**) alanine aminotransferase (ALT, U/L) and (**i**) aspartate aminotransferase (AST, U/L). The orange graphs depict the changes in renal function according to (**j**) blood urea nitrogen (BUN, mg/dL) and (**k**) creatinine (mg/dL). Significant differences (*p* < 0.05) are indicated by capital letters in the experimental group and by lowercase letters in the control group according to Tukey’s post hoc tests.

**Figure 4 animals-10-01698-f004:**
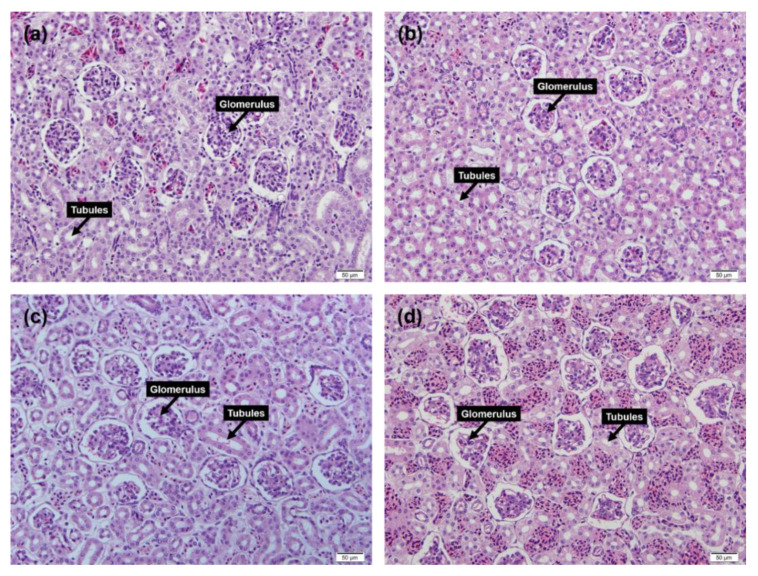
Renal tissue of experiment and control groups under both extreme and moderate conditions: (**a**) renal tissue of the experimental group under the severe condition, (**b**) renal tissue of the control group under the severe condition, (**c**) renal tissue of the experimental group under the moderate condition, and (**d**) renal tissue of control group under the moderate condition. We analyzed histological abnormalities of the glomerulus and tubules.

**Figure 5 animals-10-01698-f005:**
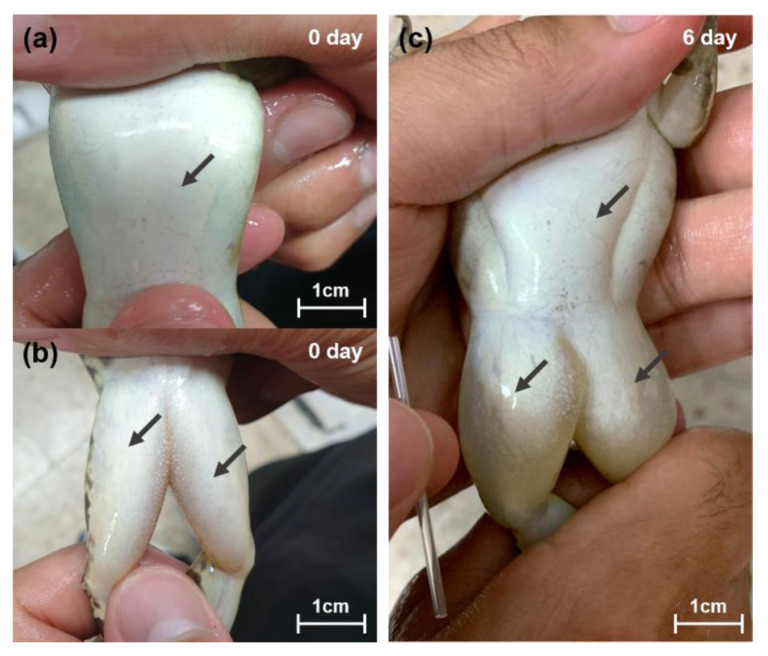
Ventral and femoral image of a frog in the severe condition: (**a**) ventral image of frog on 0 days (before salinity exposure), (**b**) femoral image of frog on 0 days (before salinity exposure), (**c**) ventral and femoral image of frog on 6 days (after salinity exposure). The individual in the image is the same frog, and after exposure to salinity, the skin in the ventral and femoral area is swollen and the ascites are full.

**Figure 6 animals-10-01698-f006:**
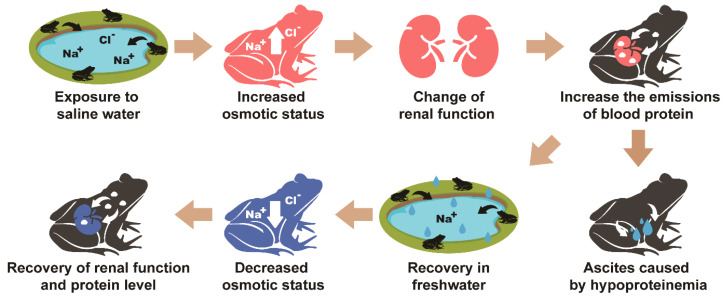
Physiological response of frogs in increased salinity of habitats and recovery from saline conditions. The red images represent increased osmotic pressure and changes in kidney function. The blue images indicate recovery from the physiological changes. When frogs are exposed to water with high osmotic potential, their renal function changes, along with their body osmotic pressure, and they have increased emissions of blood proteins. When frogs are exposed to lower osmotic water, their renal function and protein concentration return to normal after their body osmotic pressure is lowered.

**Table 1 animals-10-01698-t001:** The reference intervals of 8 electrolyte components (sodium, chlorine, and potassium) and 8 serum components (total protein, albumin, total globulin, glucose, alanine aminotransferase; ALT, aspartate aminotransferase; AST, blood urea nitrogen; BUN, and creatinine) in 34 wild-caught frogs. We analyzed the mean ± SD, 95% confidence interval (CI), median, and first to third quartile values (25% to 75%) of each component.

Components	Mean ± SD	95% CI	Median	25% to 75%
Sodium	105.6 ± 12.3	101.6–109.6	105.6	98.5–111.7
Chlorine	70.7 ± 3.2	67.5–73.9	69.4	63.9–76.2
Potassium	3.3 ± 0.8	3.00–3.5	3.1	2.6–3.6
Total protein	2.2 ± 1.1	1.8–2.6	2.07	1.5–2.7
Albumin	0.6 ± 0.3	0.47–0.66	0.53	0.37–0.66
Total globulin	1.7 ± 0.8	1.3–1.9	1.57	1.1–2.0
Glucose	25.1 ± 31.6	14.7–35.4	14.32	10.1–30.2
ALT	66.2 ± 115.1	28.6–103.8	28.63	17.5–60.1
AST	264 ± 257	180–348	200.41	127.3–306.7
BUN	24.1 ± 15.3	19.1–29.2	25.4	9.9–36.2
Creatinine	0.1 ± 0.06	0.08–0.12	0.1	0.06–0.15

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
