# Peer review of "Physiological Response of Pelophylax nigromaculatus Adults to Salinity Exposure"

_animals, 2020, doi:10.3390/ani10091698_

Round 1

Reviewer 1 Report

Response to reviewers is carefully written, and allsections are greatly improved during revision. Also, discussions have been changed at some points. I appreciate the effort of the authors

Author Response

Dear Reviewer

Thank you for your review.

Best Regards

Yuno Do

Reviewer 2 Report

Overall, the authors did respond and changed the MS according to the suggestions made by the reviewer.

Please see just a few minor comments. I do not have any further comment or doubt. 

Line 17: could be "resilience" a more fit adjective to "sustenance"?

Line 28: please consider "can be relisient" instead of "can be existed". Also line 378, 390.

Line 50: Consider susbitute various by broad

Line 84: add S in adult

Author Response

Dear Reviewer

Thank you for your comments. We changed the phrases in the text.

Best Regards

Yuno Do

This manuscript is a resubmission of an earlier submission. The following is a list of the peer review reports and author responses from that submission.

Round 1

Reviewer 1 Report

The manuscript of Park and Yuno aims to contribute to a better understanding of the toxicity of salinity in Pelophylax nigromaculatus.

They investigated the physiological responses of adults after exposure to two experimental conditions: i. "severe condition" - a short exposure (6 days) to high salinity in an aquarium during which frogs could not leave the water ii. "moderate condition" - a prolonged exposure (40 days), in a more natural situation, during which frogs had the opportunity to leave water (the salinity of the water was the same); this prolonged exposure also encompasses a recovery period (20 days). The authors also assessed the presence of kidney histological alterations.

Overall, the results of this manuscript may be interesting for "Animals" readers, but the paper needs some improvement before it would be allowed for publication. Reasonably a critical and thorough rereading/rewriting would improve the manuscript.

Below I summarize the most critical aspects (specific comments are annotated in the attached .pdf)

  1. First, English requires complete proofreading by a native speaker. I found some linguistic naivety throughout the whole Ms, and the text needs to be carefully checked (for example authors use the first capital letter after the parenthesis)
  2. Title. It would be better "Physiological response of Pelophylax nigromaculatus adults to salinity exposure."
  3. Results. I found the results section very difficult to read: authors should make an effort to rewrite more clearly, perhaps avoiding the use of abbreviations, which in this case makes reading more difficult or at least ALWAYS using them.

My major concern is related to the histological analysis; first, I could not understand what condition the images refer to, and the description is poor.

It seems that the authors do not have enough experience in histology. Still, they must make an effort to analyze the kidney tissue and describe all histological sections in an appropriate and detailed manner for all experimental conditions (and all times).

  1. Material and methods. The whole paragraph materials and methods should be completely reorganized. Some information is scattered throughout the text, and you have to find it. That makes it difficult to read the text fluently and to clearly understand the experimental setup.

Figure 1 needs to be modified: the authors wrote in the legend LINE 94 "after 60 days of recovery and control," but I cannot understand what they mean!. Moreover, the recovery period lasted 20 days: therefore, 40 days of exposure + 20 of recovery.

LINE 98 The authors wrote, "Under the severe condition, frogs experienced either salinity exposure or no exposure". This does not describe the experimental condition: when animals entered the water, they were always exposed to high salinity; the only thing that cannot be known is how long each animal remained in the water. This sentence is proposed several times in the text.

The description of histological methods is missing.

  1. Discussions remain largely speculative. The whole section needs to be rewritten entirely.

Author Response

Response to reviewers' comments: Reviewer 1

The manuscript of Park and Yuno aims to contribute to a better understanding of the toxicity of salinity in Pelophylax nigromaculatus.

They investigated the physiological responses of adults after exposure to two experimental conditions: i. "severe condition" - a short exposure (6 days) to high salinity in an aquarium during which frogs could not leave the water ii. "moderate condition" - a prolonged exposure (40 days), in a more natural situation, during which frogs had the opportunity to leave water (the salinity of the water was the same); this prolonged exposure also encompasses a recovery period (20 days). The authors also assessed the presence of kidney histological alterations.

Overall, the results of this manuscript may be interesting for "Animals" readers, but the paper needs some improvement before it would be allowed for publication. Reasonably a critical and thorough rereading/rewriting would improve the manuscript.

→ We appreciate reviewer’s positive view on our manuscripts. We tried our best to reflect the reviewer’s comments during the revision.

Below I summarize the most critical aspects (specific comments are annotated in the attached .pdf)

  1. First, English requires complete proofreading by a native speaker. I found some linguistic naivety throughout the whole Ms, and the text needs to be carefully checked (for example authors use the first capital letter after the parenthesis)

→ We have been completely proofreading by native speakers. We have corrected all additional explanations after parentheses to lowercase (except equipment names, software name, scientific name, and abbreviations etc.). Also, the overall English was checked again, including the checked sentences and words in the review file.

  1. Title. It would be better "Physiological response of Pelophylax nigromaculatusadults to salinity exposure."

→ We agree your comment. The title was revised.

  1. Results. I found the results section very difficult to read: authors should make an effort to rewrite more clearly, perhaps avoiding the use of abbreviations, which in this case makes reading more difficult or at least ALWAYS using them.

→ We overall revised the results section. The results of the experimental group and the control group were separated, and integrated for each group. Also, all abbreviations were corrected in all section, except AST, ALT, and BUN, which are commonly used as abbreviations in clinical analysis.

My major concern is related to the histological analysis; first, I could not understand what condition the images refer to, and the description is poor.

It seems that the authors do not have enough experience in histology. Still, they must make an effort to analyze the kidney tissue and describe all histological sections in an appropriate and detailed manner for all experimental conditions (and all times).

→ We have re-written the process of making the sample to the process of performing the test in the histological analysis method.

Page5 Line159-182: After the experimental period was over, six frogs from server conditions (3 frogs of experimental group and 3 frogs of control group) and ten frogs from moderate conditions (5 frogs of experimental groups and 5 frogs of control groups) were euthanized by pithing, and renal tissue of frogs were rapidly collected and fixed in 10% formalin solution. A fixed sample of renal tissue was cut to a thickness of 2-3 mm, and was washed in deionized water. The paraffin embedding of samples was carried out by using Spin Tissue Processor STP 120 (Myr, Tarragona, Spain) into three steps: (1) Dehydration process using ethanol (2) clearing process using xylene (3) paraffin embedding process. Thermo-Shandon Finesse ME Microtome (Thermo Fisher Scientific, Waltham, USA) were used to make the tissue section created by cutting to 3µL thickness. The resulting tissue section was attached to a slide glass, and washed with deionized water after section was dried, was deparaffinized, and was hydration process. Afterwards, the following Harris's hematoxylin and eosin staining process was performed: (1) The samples were stained with hematoxylin for 10 minutes, and the stained samples were washed with running water for 3 minutes (2) after staining the samples with the eosin solution for 1 minute and 40 seconds, a 4-step hydration process was performed for 1 minute (3) Clearing was performed for 3 minutes using xylene, and after fixing the samples using the glycerol, the cover glass was covered and enclosed. The glomerulus and tubules of the renal system were examined to identify renal failure, and abnormalities due to saline exposure were examined using a light microscope with a camera (Olympus DP72, Japan) at 100x magnification. In the case of the glomerulus, a predominance of polymorphonuclear leukocytes and glomerular congestion are observed when there is a histological renal dysfunction. It can also appear as patches and expansion of the glomerular mesangium. In tubules with abnormal renal function, the reactive and degenerative nuclear changes, intraluminal eosinophilic material, remaining nuclei with prominent nucleolus, acute tubular necrosis etc. can be observed, represented in the shape of the nucleus or deformation of the tubular edge. It also develops a tumor form in which the meninges of the tubules swell [33]. We confirmed the histological renal dysfunction of treatment groups from severe and moderate conditions through comparison with the renal image of control groups from severe and moderate conditions.

Page8 Line266-273: When the three renal image of the treatment group in severe group were compared with the three renal image of the control group, evidence of histological renal abnormalities (such as patches and expansion of the glomerular mesangium, reactive and degenerative nuclear changes, intraluminal eosinophilic material, remaining nuclei with prominent nucleolus , acute tubular necrosis, deformation of the tubular edge etc.) were not found in the renal image of frogs in treatment group (Figure 4a, 4b). Under the moderate condition, the five renal image of treatment groups was not represented as histological renal abnormalities when the renal images were compared the five image of control groups (Figure 4c, 4d).

  1. Material and methods. The whole paragraph materials and methods should be completely reorganized. Some information is scattered throughout the text, and you have to find it. That makes it difficult to read the text fluently and to clearly understand the experimental setup.

 → We have modified the sentence and sequence in material and methods section, and especially reconstructed and modified the section of 2.2. Experimental design and treatment.

Figure 1 needs to be modified: the authors wrote in the legend LINE 94 "after 60 days of recovery and control," but I cannot understand what they mean! Moreover, the recovery period lasted 20 days: therefore, 40 days of exposure + 20 of recovery.

 → We revised this sentence.

Page3 Line94 at last scheme: After 40-days of exposure + 20-days of recovery and 60-days of control

LINE 98 The authors wrote, "Under the severe condition, frogs experienced either salinity exposure or no exposure". This does not describe the experimental condition: when animals entered the water, they were always exposed to high salinity; the only thing that cannot be known is how long each animal remained in the water. This sentence is proposed several times in the text.

 → We revised these sentences.

Page3 Line97-107: We set up two experimental conditions: a severe and a moderate exposure (Figure 1. In the first case (severe condition), frogs were bred in closed rectangular plastic container (460 mm × 300 mm × 170 mm) with a water pool but no bed of fiber. Under the severe condition, frogs remained in the water during the entire experimental period and could not escape the water. Six frogs were divided into experimental group (three individuals) and control group (three individuals). Frogs in the experimental groups were exposed to 5‰ saline water for six days, while frogs in the control groups were maintained to freshwater below 0.4 ‰ salinity for six days.

Page4 Line113-124: In the second case (moderate condition), frogs were bred in plastic container (460 mm × 300 mm × 170 mm) with a water pool and the bed of fiber. Frogs had free access to land (on the bed of fiber) and water (in the pool) in moderate conditions. 10 frogs were divided into experimental group (five individuals) and control group (five individuals). Frogs of experimental group were exposed to 5‰ saline water for 40 days (the salinity treatment period). After salinity exposure, frogs were kept in a container with freshwater less than 0.4‰ salinity for 20 days (the recovery period). Other five frogs in the control group were raised with less than 0.4 ‰ of bottled water for 60 days.

The description of histological methods is missing.

 → We have described the detail process of making the sample to the process of performing the test in the histological analysis method.

Page5 Line159-182: After the experimental period was over, six frogs from server conditions (3 frogs of experimental group and 3 frogs of control group) and ten frogs from moderate conditions (5 frogs of experimental groups and 5 frogs of control groups) were euthanized by pithing, and renal tissue of frogs were rapidly collected and fixed in 10% formalin solution. A fixed sample of renal tissue was cut to a thickness of 2-3 mm, and was washed in deionized water. The paraffin embedding of samples was carried out by using Spin Tissue Processor STP 120 (Myr, Tarragona, Spain) into three steps: (1) Dehydration process using ethanol (2) clearing process using xylene (3) paraffin embedding process. Thermo-Shandon Finesse ME Microtome (Thermo Fisher Scientific, Waltham, USA) were used to make the tissue section created by cutting to 3µL thickness. The resulting tissue section was attached to a slide glass, and washed with deionized water after section was dried, was deparaffinized, and was hydration process. Afterwards, the following Harris's hematoxylin and eosin staining process was performed: (1) The samples were stained with hematoxylin for 10 minutes, and the stained samples were washed with running water for 3 minutes (2) after staining the samples with the eosin solution for 1 minute and 40 seconds, a 4-step hydration process was performed for 1 minute (3) Clearing was performed for 3 minutes using xylene, and after fixing the samples using the glycerol, the cover glass was covered and enclosed. The glomerulus and tubules of the renal system were examined to identify renal failure, and abnormalities due to saline exposure were examined using a light microscope with a camera (Olympus DP72, Japan) at 100x magnification. In the case of the glomerulus, a predominance of polymorphonuclear leukocytes and glomerular congestion are observed when there is a histological renal dysfunction. It can also appear as patches and expansion of the glomerular mesangium. In tubules with abnormal renal function, the reactive and degenerative nuclear changes, intraluminal eosinophilic material, remaining nuclei with prominent nucleolus, acute tubular necrosis etc. can be observed, represented in the shape of the nucleus or deformation of the tubular edge. It also develops a tumor form in which the meninges of the tubules swell [33]. We comfirmed the histological renal dysfunction of treatment groups from severe and moderate conditions through comparison with the renal image of control groups from severe and moderate conditions.

  1. Discussions remain largely speculative. The whole section needs to be rewritten entirely.

 → We revised the part of most discussions. We tried to reconstruct the paragraph, sentence, and word. Also, some speculative sentences were corrected or deleted in part of discussions.

Reviewer 2 Report

Review assignment on: Animals-885448-peer-review-v1

Overall, I do find this research paper very interesting, raising some very interesting concepts, namely on the concept of avoidance (here simulated by the moderate condition scenario).

The schematic figures do help to follow the proceedings of the work.

I also think that is very valuable as research studies in amphibian’s species are still very scarce despite amphibians are one of the most endangered groups of vertebrates. Moreover, it focused on adults and authors have performed an extensive battery of metabolic endpoints.

Major comments:

Line 34: references. I would avoid mixing soil salinization with freshwater salinization.

You can check others:

  • “Sensitivity of freshwater species under single and multigeneration exposure to seawater intrusion”
  • “Sensitivity to salinization and acclimation potential of amphibian (Pelophylax perezi) and fish (Lepomis gibbosus) models”

Amphibians are one (if not the most) diverse group of vertebrates, but they are also highly threatened, experiencing the highest extinction rates due to a wide range of factors. Salinity is one of those factors. But I think that this framework is missing and should be added as it increases the value and scope of the article. Maybe you can check: “Validity of fish, birds and mammals as surrogates for amphibians and reptiles in pesticide toxicity assessment and references therein for this framework on amphibian’s decline.”

I do have some reserves in using the term “adaptation”. The use of the term implies that some genetic alterations might have occurred, which was not studied in the present MS. Since the authors verified that the serum components returned to their basal state after a period of saline stress, I would considered that it is possibly related to tolerance mechanisms, which would also go inline with the “physiological resilience” term used in the title.   Please see throughout the manuscript.

In order to broaden the scope of your paper and later, their probability to be find by other researchers, I would suggest the authors to give (alongside the salinity values that they already use) a proxy in conductivity. Salinity is not all the same, salt composition may vary along the position on the globe, or even the source of pollution (for instance, salt composition varies a lot when talking about road deicing salts). I think that a proxy value in conductivity along your salinity value would also help other researchers to interpret the level of exposure.

Figure 2. Some endpoints do need a little more focus (on the possible why) in the discussion section. For instance, why did alanine aminotransferase levels (despite not statistically meaning) decrease in the control group, as well as albumin and total protein? Can be due to lack of food? Maybe due to handling stress?

Minor comments:

Line 12: 6 instead of six

Line 13, 23: Please specify whether the authors are referring to severe or moderate salinity condition. Please see also where needed in the remaining manuscript.

Line 81: acclimation occurred for how long? Please indicate.

Line 94: legend of the last scheme on the moderate exposure scenario. Instead of “60 days of recovery” shouldn’t be “20 d of recovery” or “after 40 d exposure + 20 d of recovery”

Section 2.2: Why 6 days for severe conditions and 40+20 days for moderate? Was it based in other works? Based on preliminary assays? Please indicate accordingly.

Section 3.1 and 3.2: I would start by mentioning control, and then the condition, i.e, switch paragraph 168-173 and 160-167.   

Author Response

Response to reviewers' comments: Reviewer 2

Overall, I do find this research paper very interesting, raising some very interesting concepts, namely on the concept of avoidance (here simulated by the moderate condition scenario).

The schematic figures do help to follow the proceedings of the work.

I also think that is very valuable as research studies in amphibian’s species are still very scarce despite amphibians are one of the most endangered groups of vertebrates. Moreover, it focused on adults and authors have performed an extensive battery of metabolic endpoints.

→ Thank you for the reviewer’s comments. We revised the sentence and word in several times. We tried our best to reflect the reviewer’s comments during the revision.

Major comments:

Line 34: references. I would avoid mixing soil salinization with freshwater salinization.

→ We deleted some of the references and correct the sentence.

Page1 Line34-35: Freshwater ecosystems have been degraded by increase of salinity [1], and many organisms inhabiting in freshwater habitats are physiologically affected by this salinization [2].

You can check others:

“Sensitivity of freshwater species under single and multigeneration exposure to seawater intrusion”

“Sensitivity to salinization and acclimation potential of amphibian (Pelophylax perezi) and fish (Lepomis gibbosus) models”

 → We added references and sentences.

Page1 Line35-36: In particular, amphibians are more sensitive to change of salinity than other taxa [3] because they have permeable skin and a complex life history [4].

Amphibians are one (if not the most) diverse group of vertebrates, but they are also highly threatened, experiencing the highest extinction rates due to a wide range of factors. Salinity is one of those factors. But I think that this framework is missing and should be added as it increases the value and scope of the article. Maybe you can check: “Validity of fish, birds and mammals as surrogates for amphibians and reptiles in pesticide toxicity assessment and references therein for this framework on amphibian’s decline.”

 → We added references and sentences.

Page1 Line37-41: Amphibians are highly threatened, experiencing the highest extinction rates in vertebrates (ca. 40%) due to various factors, but most of these factors (ca. 48%) are "enigmatic factors" [5]. The enigmatic threaten factors of amphibians may include change of salinity in freshwater habitats. The changes of salinity have various effects on the amphibian population [6-8], and can actually be frequently exposed in them.

Page2 Line46-49: We need to understand the physiological response of amphibians in all life history by change of salinity to come up with ways to efficiently protect and manage them and to identify how they survive and adapt in predictable or unpredictable saline habitats.

I do have some reserves in using the term “adaptation”. The use of the term implies that some genetic alterations might have occurred, which was not studied in the present MS. Since the authors verified that the serum components returned to their basal state after a period of saline stress, I would consider that it is possibly related to tolerance mechanisms, which would also go in line with the “physiological resilience” term used in the title. Please see throughout the manuscript.

 → We agree the opinion of reviewer. We corrected the all statement that our results suggest adaptation.

Page1 Line17-18: Adult anurans seem capable of sustenance, to some extent, to saline conditions.

Page1 Line28-29: Thus, adult anurans can be existed, to some extent, to saline conditions in habitats that experience either rapid or slow salinity changes.

Page9 Line291-292 at Figure 6 legend: Physiological response of frogs in increased salinity of habitats and recovery from saline conditions.

Page11 Line377-378: As a result, adult frogs can be existed to saline habitats, as shown by their ability to recover normal functioning when saline conditions were recovered.

In order to broaden the scope of your paper and later, their probability to be find by other researchers, I would suggest the authors to give (alongside the salinity values that they already use) a proxy in conductivity. Salinity is not all the same, salt composition may vary along the position on the globe, or even the source of pollution (for instance, salt composition varies a lot when talking about road deicing salts). I think that a proxy value in conductivity along your salinity value would also help other researchers to interpret the level of exposure.

 → We added the contents of conductivity in material and methods section.

Figure 2. Some endpoints do need a little more focus (on the possible why) in the discussion section. For instance, why did alanine aminotransferase levels (despite not statistically meaning) decrease in the control group, as well as albumin and total protein? Can be due to lack of food? Maybe due to handling stress?

 → We agree your comments. We explained most serum components in the discussion part, except ALT and AST. In particular, in the salinity exposure + recovery experiment, we were able to argue that this was not due to handling stress or lack of food because the change patterns of serum components were checked during exposure and recovery. In severe conditions, although ALT tends to decrease, it is not likely due to lack of food or stress. In process of converted protein into glucose in the liver, the levels of BUN, glucose, and protein were changed, and AST (despite not statistically meaning) seems to have changed during this process. In addition, the ALT level (despite not statistically meaning) gradually decreased in the control, which seems to have occurred as it gradually stabilized in container without stress. Descriptions of all changing contents except those with no statistical difference are included in the discussion part.

Minor comments:

Line 12: 6 instead of six

→ We revised this word.

Page1 Line11-13: Analysis of 11 serum components revealed the physiological response of frogs to either severe saline exposure for six days or moderate saline exposure for 40-days, followed by a 20-days recovery period.

Line 13, 23: Please specify whether the authors are referring to severe or moderate salinity condition. Please see also where needed in the remaining manuscript.

→ We had added an explanation for this part.

Line 81: acclimation occurred for how long? Please indicate.

→ We added about this content.

Page2 Line89-93: In the lab, frogs were acclimated during 72-hr to the study conditions in covered plastic aquaria (460 mm × 300 mm × 170 mm) with a bed of sterile coconut fiber and free access to a water pool containing 2 L of bottled water under a 12-hr/12-hr light/dark regime with Exo-tera UVB 100 lamps (UVB 30%, UVA 5%) and air temperature 25 ±â€Ż2  °C.

Line 94: legend of the last scheme on the moderate exposure scenario. Instead of “60 days of recovery” shouldn’t be “20 d of recovery” or “after 40 d exposure + 20 d of recovery”

 → We revised this sentence.

Page3 Line94 at last scheme: After 40-days of exposure + 20-days of recovery and 60-days of control

Section 2.2: Why 6 days for severe conditions and 40+20 days for moderate? Was it based in other works? Based on preliminary assays? Please indicate accordingly.

 → We designed the experiment based on preliminary analysis. This content has been added to the text.

Page3 Line101-103: In a preliminary test, the serum sodium and chlorine of frogs under severe conditions were significantly changed on 3 days to 6 days, and changed serum components were maintained since then

Section 3.1 and 3.2: I would start by mentioning control, and then the condition, i.e, switch paragraph 168-173 and 160-167.   

→ We have reorganized the sentences and sequence according to the reviewer's comments. The results of the experimental group and the control group were separated, and integrated for each group.
